# IGF-1 Haploinsufficiency Causes Age-Related Chronic Cochlear Inflammation and Increases Noise-Induced Hearing Loss

**DOI:** 10.3390/cells10071686

**Published:** 2021-07-03

**Authors:** Adelaida M. Celaya, Lourdes Rodríguez-de la Rosa, Jose M. Bermúdez-Muñoz, José M. Zubeldia, Carlos Romá-Mateo, Carlos Avendaño, Federico V. Pallardó, Isabel Varela-Nieto

**Affiliations:** 1Institute for Biomedical Research “Alberto Sols” (IIBM), Spanish National Research Council-Autonomous University of Madrid (CSIC-UAM), 28029 Madrid, Spain; acelaya@iib.uam.es (A.M.C.); jmbermudez@iib.uam.es (J.M.B.-M.); josemanuel.zubeldia@salud.madrid.org (J.M.Z.); 2Rare Diseases Biomedical Research Networking Centre (CIBERER), The Institute of Health Carlos III (ISCIII), 28029 Madrid, Spain; Carlos.Roma@uv.es (C.R.-M.); federico.v.pallardo@uv.es (F.V.P.); 3Hospital La Paz Institute for Health Research (IdiPAZ), 28029 Madrid, Spain; carlos.avendano@uam.es; 4Allergy Service, Gregorio Marañon General University Hospital, 28009 Madrid, Spain; 5Gregorio Marañon Health Research Institute (IiSGM), 28009 Madrid, Spain; 6Department of Physiology, Faculty of Medicine and Dentistry, University of Valencia, Spain and FIHCUV-INCLIVA, 46010 Valencia, Spain; 7Department of Anatomy, Histology & Neuroscience, Medical School, Autonomous University of Madrid, 28029 Madrid, Spain

**Keywords:** AKT, apoptosis, ARHL, IL1β, JNK, NIHL, TGFβ1

## Abstract

Insulin-like growth factor 1 (IGF-1) deficiency is an ultrarare syndromic human sensorineural deafness. Accordingly, IGF-1 is essential for the postnatal maturation of the cochlea and the correct wiring of hearing in mice. Less severe decreases in human IGF-1 levels have been associated with other hearing loss rare genetic syndromes, as well as with age-related hearing loss (ARHL). However, the underlying mechanisms linking IGF-1 haploinsufficiency with auditory pathology and ARHL have not been studied. *Igf1*-heterozygous mice express less *Igf1* transcription and have 40% lower IGF-1 serum levels than wild-type mice. Along with ageing, IGF-1 levels decreased concomitantly with the increased expression of inflammatory cytokines, *Tgfb1* and *Il1b*, but there was no associated hearing loss. However, noise exposure of these mice caused increased injury to sensory hair cells and irreversible hearing loss. Concomitantly, there was a significant alteration in the expression ratio of pro- and anti-inflammatory cytokines in *Igf1*^+/−^ mice. Unbalanced inflammation led to the activation of the stress kinase JNK and the failure to activate AKT. Our data show that IGF-1 haploinsufficiency causes a chronic subclinical proinflammatory age-associated state and, consequently, greater susceptibility to stressors. This work provides the molecular bases to further understand hearing disorders linked to IGF-1 deficiency.

## 1. Introduction

Age-related hearing loss (ARHL) is a disabling pathology that affects one third of the population over 65 years. ARHL is influenced by multiple factors including infectious diseases, ototoxic drugs, and exposure to excessive noise [1].There is also a genetic predisposition to suffer ARHL, although the genes involved are not yet fully identified [2,3].

Human insulin-like growth factor 1 (IGF-1) homozygous deficiency is an ultrarare disease associated with dwarfism, mental retardation, and syndromic sensorineural hearing loss (SNHL) [4]. However, IGF-1 heterozygous mutations have not been associated with congenital SNHL yet [5,6,7]. Interestingly, rare human illnesses such as Laron and Turner syndromes show a close association between low IGF-1 serum levels and progressive SNHL [8,9,10,11]. Circulating IGF-1 levels in mammals undergo a physiological age-related decrease [12,13] that has been associated with cognitive decline and neurodegeneration [14,15], as well as with SNHL [11].

Noise plays a central role among environmental factors contributing to the progression of ARHL [16,17]. Oxidative stress and inflammation are early molecular mechanisms of response shared by noise injury and ARHL progression [18]. Local cytokine secretion attracts inflammatory cells to clear cell debris, but these cells may eventually spread cell injury and ultimately lead to cell death and SNHL [19,20]. In this context, IGF-1 is otoprotective in models of noise-induced hearing loss (NIHL) [21,22,23] and has been shown to maintain the cochlear ribbon synapse ex vivo [24]. IGF-1 modulates neuroinflammation [25] and regulates immune cells functions; in turn, its expression and signaling are regulated by the products of the inflammatory milieu [26,27]. Interestingly, IGF-1 has been shown to be otoprotective in human sudden HL, another otic inflammatory imbalance and oxidative stress condition [28,29,30]. Though, the molecular mechanisms linking IGF-1 haploinsufficiency with increased susceptibility to SNHL of multiple aetiologies have not yet been studied.

Here, we showed that IGF-1 levels are downregulated with ageing and that IGF-1 haploinsufficiency induces an age-related subclinical chronic inflammatory state in the mouse cochlea. *Igf1*^+/−^ mice did not show premature or accelerated ARHL, but, when exposed to noise, showed a more severe NIHL than wild-type mice. Cochleae of *Igf1*^+/−^ mice showed an exacerbated extent of noise-induced injury that included increased inflammation, oxidative stress, and apoptosis. Interestingly, the impact of IGF-1 deficiency on cochlear noise susceptibility was age-dependent.

Taken together, our data strongly indicate that IGF-1 age-related downregulation is a pro-inflammatory condition that contributes to the pathogenesis of ARHL. We propose here that IGF-1 circulating levels have a threshold under which there is a less effective cochlear control of the inflammatory response and survival signaling.

## 2. Materials and Methods

Mouse handling and genotyping. MF1/129SvEvTac *Igf1*^+/−^ mice [31] were genotyped [32] and bred to obtain *Igf1*^+/−^ and *Igf1*^+/+^ mice (WT). No differences among male and female mice were observed.

Hearing evaluation and noise exposure. Auditory brainstem responses (ABRs) and distortion product otoacoustic emissions (DPOAEs) were recorded with a Tucker Davis Technologies workstation (TDT) [33]. Briefly, for the ABR test, click and tone burst stimuli (8, 16, 20, and 40 kHz) were presented with an MF1 magnetic speaker (TDT) from 90 to 20 dB SPL in 5–10 dB SPL steps. Click stimuli were 0.1 ms and tone burst stimuli were 5 ms in duration (2.5 ms each for rise and decay, without plateau). The threshold of click-evoked and tone-evoked ABRs, peak latencies, and amplitudes were determined. For DPOAE, an ER10B+ probe (Etymotic Research Inc., IL, USA) was inserted into the external auditory canal, and mice were stimulated with two synchronic tones, whose frequencies (f1, f2; relation f1/f2 = 1.2) were calculated from a central frequency (*F* = 8 and 10 kHz; f1 = *F* × 0.909, f2 = *F* × 1.09), and presented with decreasing intensity from 80 to 30 dB SPL (f1 level = f2 level).The distortion product 2f1–f2 was determined for each sound level from the FFT waveforms. DPOAE thresholds were defined as the minimum level of the primary tones that elicit a 2f1–f2 response higher than background noise [34]. Four *Igf1*^+/−^ and 3 WT 1-month-old (young) mice and 13 *Igf1*^+/−^ and 11 WT 6-month-old (adult) mice were tested. Data analysis was performed with BioSigRP TM software (TDT). Mice were exposed to a violet swept sine noise (VSSN, frequency range 2–20 kHz) at 110 dB SPL for 30 min, and hearing was evaluated by ABR before and 1 h and 3, 14, and 28 days after noise exposure [34,35]. Briefly, conscious mice were confined in a wire mesh cage in the center of a reverberant chamber acoustically designed to reach the maximum sound level with minimum deviation in the central exposure area and exposed to violet swept sine (VS) noise, at 100–120 dB SPL for 30 min as reported [35]. VS noise was repeated during the 30 min of exposure. VS noise was designed with Wavelab Lite software (Steinberg Media Technologies GmbH, Hamburg, Germany). It consists in a 10 s linear sweep in frequency, with a spectrum biased towards high frequencies (frequency range 2–20 kHz) and presented with a linear-with-frequency gain [19,35,36,37].

Cochlear morphology and immunohistochemistry. For cochlear histological evaluation, paraffin sections (5 μm) or cryosections (10 μm) were processed for Nissl or hematoxylin-eosin staining [32,38]. Mice were anesthetized with pentobarbital (Dolethal, Bayer, 150 mg/kg) and perfused transcardially with 4% paraformaldehyde in 0.1 M phosphate-buffered saline (PBS) (pH 7.4). Cochleae were dissected, fixed overnight in 4% paraformaldehyde in PBS (4 °C), decalcified in 0.3 M EDTA (pH 6.5) for 8 days, and embedded in paraffin or gelatine. Paraffin sections (10 μm) or cryosections (20 μm) of cochleae from WT and *Igf1*^+/−^ mice were stained with haematoxylin-eosin to study cochlear cytoarchitecture [34,39]. For immunofluorescence assays, serial cryostat sections were incubated overnight with primary antibodies (Appendix A) and then with Alexa-Fluor-conjugated secondary antibodies for 2 h (RT). Sections were mounted in Prolong Gold containing DAPI (Invitrogen) and visualized using a fluorescence microscope (Nikon 90i, Tokyo, Japan) [37]. Digital images were obtained by epifluorescence microscopy (Nikon 90i) using a DS-Qi1Mc camera and NisElements 3.01 software. Synaptophysin, neurofilament, and IBA1 intensities were determined in the middle turn of 4 equivalent sections prepared from at least 3 mice per condition. Quantification was performed using ImageJ software (National Institutes of Health) [40].

Organ of Corti dissection and hair cell quantification. Cochleae were post-fixed and decalcified in 0.3 M EDTA and then dissected under a stereomicroscope (Nikon C-DSD230). Osseous and membranous labyrinths were removed, and cochleae were sectioned to separate the apex and middle turns from the basal turn. Samples were treated with PBS with 0.5% Triton X-100, incubated for 1 h at RT with Alexa Fluor 488 phalloidin (Molecular Probes, Eugene, OR, USA) at 1:100 in PBS with 0.5% Triton X-100, and placed in 8-well microscope slides (Menzel-Gläser) with Vectashield/DAPI mounting medium. Hair cell (HC) density was estimated using a stereological approach. The area of interest containing the rows of HC was divided into equidistant 5% sectors using CAST^®^ stereological software (v.2.3.2.0, Visiopharm; Hoersholm, Denmark) in an Olympus BX51 microscope connected to an Olympus DP70 video camera with a source of ultraviolet light (Olympus U-RFL-T; Olympus, Tokyo, Japan). HC numbers were determined on systematic randomly sampled areas from each sector using unbiased counting frames. Sets of 12 closely spaced frames were superimposed on every sampled area and phalloidin stained inner (IHC) and outer (OHC) hair cells with intact hair-bundles were counted as previously reported [34]. Briefly, each HC is unequivocally represented by its hair bundle, phalloidin-stained stereocilia that were used as counting units, and HC were counted as present if the hair bundle was intact. The HC count was carried out by placing unbiased frames in a uniformly random sampling strategy on the focus planes of the cilia. The reference space was considered the stereociliary fringe, that is, the convex hull area bounding the region containing the rows of hair tufts of the HCs. The IHC and OHC cell densities were estimated for each sector as follows: N_A_ (HC) = ∑Q (HC) ×1000/∑a (SF), where N_A_ is the cell density expressed as the number of cells/1000 µm^2^, ∑Q is the sum of HC counted within each sector, and ∑a (SF) is the area of interest, in µm^2^, sampled with the unbiased frames within the given sector. The precision of this method was approximated by computing the coefficient of error (CE) of the estimates (N_A_) obtained on each sector, applying Cochran’s equation for ratio estimators (eq. 10.32 as reported in [41]). Our strategy yielded CE values of ~17% (0.17 ± 0.015) in the control animals and ~21% (0.21 ± 0.020) in the noise-damaged groups. Cytocochleograms were constructed by plotting the number of present OHCs or IHCs, or the OHC and IHC densities, as a function of percent distance from the apex of the cochlea [34].

TUNEL staining. TdT-mediated dUTP nick-end labelling (TUNEL) of fragmented DNA was visualized using the ApopTag kit (Merck-Millipore, Danvers MA, USA). TUNEL-positive nuclei were examined in 4–12 sections from 3–5 mice of each condition under an Axiophot Zeiss microscope equipped with an Olympus DP70 digital camera (Olympus).

Western blotting. Cochleae were dissected and stored at −80 °C until analysis. Cochlear proteins were extracted, resolved using SDS-PAGE electrophoresis (Mini-PROTEAN^®^ TGX™, Bio-Rad) and transferred to PVDF membranes (0.2 μm). Membranes were blocked and incubated overnight at 4 °C with primary antibodies (Appendix A) and then with peroxidase-conjugated secondary antibodies (Bio-Rad, Hercules, CA, USA) at RT for 1 h [37]. Immunoreactive bands were visualized using Clarity^TM^ Western ECL Substrate kit (Bio-Rad) in an ImageQuant LAS 4000 mini apparatus and quantified by densitometry with ImageQuant™ TL software (GE Healthcare Bio-Sciences, Chicago, IL, USA).

Gene expression. RNA was extracted from frozen cochleae using RNeasy kit (Qiagen) and analyzed with an Agilent Bioanalyzer 2100 (Agilent Technologies, Santa Clara, CA, USA). RT-qPCR was carried out with QuantiTect Primer assays (Qiagen) and Power SYBR Green PCR Master Mix or with TaqMan Gene Expression Assays (Applied Biosystems). Probes used included: *Mat2a* (QT00253372, Qiagen), *Ahcy* (QT00171612), *Gclc* (QT00130543), *Gclm* (QT00174300), *Igf1* (Mm00439561_m1, Applied Biosystems), *Igf2* (Mm00439564_m1), *Igf1r* (Mm00802831_m1), *Igbp2* (Mm00492632_m1), *Tgfb1* (Mm01178820_m1), *Tgfb2* (Mm00436955_m1), *Tgfbr1* (Mm00436964_m1), *Tgfbr2* (Mm00436977_m1), *Il1b* (Mm00434228_m1), *Il6* (Mm00446190_m1), *Tnfa* (Mm99999068_m1), *Il10* (Mm00439614_m1), *Dusp1* (Mm00457274_g1), *Gap43* (Mm00500404_m1), *Ntn1* (Mm00500896_m1), and *Foxp3* (Mm00475162_m1). Relative quantification values were calculated as 2^−ΔΔCt^ (RQ) [42] using *Rplp0* and *Gapdh* as reference genes [35].

Determination of serum IGF-1 concentration. Blood was extracted by cardiac puncture. Tubes were centrifuged at 2500 rpm for 15 min at 4 °C. The serum supernatant was stored at −80 °C until use. The concentration of IGF-1 was determined by ELISA (OCTEIA Rat/Mouse IGF-1 kit, IDS Ltd., Boldon, UK) according to the manufacturer’s protocol.

Statistical analysis. Statistical analysis was performed using SPSS v21.0 software. For ABR data, a linear mixed model procedure was performed. Statistical significance between groups was estimated by Student’s *t*-test after using Levene’s test to confirm the homogeneity of variances. RT-qPCR data were evaluated by the non-parametric Mann–Whitney U test unless otherwise stated. Data are expressed as mean ± SEM. Results were considered significant at *p* < 0.05.

## 3. Results

### 3.1. Comparative IGF-System and Cytokines Gene Expression Profiling, IGF-1 Serum Levels, and Hearing Thresholds of Igf1^+/−^ and WT Mice along Age

Temporal cochlear gene expression profiles were studied in WT and *Igf1*^+/−^ mice. Expression of all IGF-system genes was high during development and decreased dramatically by postnatal day (P) 15. Cochlear *Igf1* expression was lower in *Igf1*^+/−^ mice at every age studied, whereas other IGF system elements showed increased perinatal expression but similar profiles in both genotypes (Figure 1A). No significant differences were observed between genotypes in the cochlear expression of *Igfbp3*, *Insr*, *Irs1*, and *Irs2*. In parallel, the expression of pro- and anti-inflammatory cytokines involved in cochlear inflammatory responses showed an age-related increase in the cochlea of both genotypes, although *Il1b* expression increased in the *Igf1*^+/−^ mice from E18.5 (Figure 1B). *Il10*, *Il6*, and *Foxp3* transcription profiles were similar in both genotypes until the age of 12 months, at which time WT cochlea showed increased expression levels (Figure 1B), suggesting an increased capacity to buffer inflammation.

Systemic serum IGF-1 levels were measured in parallel and a decrease with age was observed in both WT and *Igf1*^+/−^ mice. IGF-1 levels were lower in *Igf1*^+/−^ than in WT mice, with significant differences appearing from 3 months of age onwards (Figure 1C, left). IGF-1 haploinsufficiency had no evident impact on hearing thresholds that were similar for both genotypes between 3–6 months of age (Figure 1C, right).

### 3.2. Adult Igf1^+/−^ Mice Show Increased Susceptibility to Noise Injury

Measurement of hearing acuity in both genotypes indicated that, at the ages studied, mice showed similar hearing thresholds. Similarly, 1 h after the noise challenge, young mice (Figure 2A, upper panel) of both genotypes presented similar 40 dB threshold shifts and recovery profiles. In striking contrast, adult *Igf1*^+/−^ mice were severely damaged 3 days after noise exposure, especially at frequencies over 8 kHz (Figure 2A, lower panel). WT mice hearing recovery began by day 14, whereas *Igf1*^+/−^ mice thresholds worsened, and 28 days after the noise challenge, differences were maintained. Wave latencies transitorily increased 3 days after noise exposure in *Igf1*^+/−^ mice (Figure 2B and Appendix A), and interpeak latencies I-II and II-IV were also transitorily increased (Appendix A). No noise-related changes were detected in IGF-1 serum levels throughout the study (data not shown).

Noise caused the loss of OHC (Figure 3) and apical fibrocytes of the spiral limbus (not shown) in both genotypes, although noise-exposed *Igf1*^+/−^ mice showed the most significant OHC loss (Figure 3A(a–f),B). Neurofilaments and synaptophysin were used to visualize nerve fibers and efferent synaptic terminals at the HC base. Both markers have been reported to be altered in the *Igf1*-null mouse [38,43]. Noise-exposed *Igf1*^+/−^ mice showed significantly less synaptophysin (Figure 3A(g–h),C). *Gap43* showed constitutively lower levels in *Igf1*^+/−^ mice and was up-regulated (2.3-fold) after the noise challenge in these mice (Figure 3D). DPOAE registered 28 days after noise challenge confirmed the increased injury in the *Igf1*^+/−^ OHC (Figure 3E).

Next, IGF-1 signaling pathways were studied at different times following the noise challenge. After 4 h, only *Igf1*^+/−^ mice showed a two-fold increase in p-ERK, whereas a 1.5-fold increment in p-AKT occurred in both genotypes (Figure 3F). Twenty-eight days later, AKT showed less activation (0.3-fold) in *Igf1*^+/−^ compared to WT mice, whereas ERK phosphorylation was similar to basal levels in both genotypes (Figure 3F).

### 3.3. Age-Related and Noise-Induced Cochlear Antioxidant Gene Expression Profiling of Igf1^+/−^ and WT Mice

Cochlear temporal expression profiles of oxidative stress-related genes implicated in cochlear homeostasis [36] were studied in WT and *Igf1*^+/−^ mice (Figure 4A; pathways represented in Figure 4B). No evident differences were observed between genotypes in methionine cycle and transsulfuration pathway genes *Mat2a*, *Ahcy*, and *Gclc* that were expressed during embryonic stages and decreased postnatally. *Gclm* transcripts increased with age. In contrast, following the noise challenge, *Mat2a* and *Ahcy* showed a peak of expression 4 h after injury only in WT mice, whereas *Gclc* and *Gclm* levels increased 3 days after noise in both genotypes but only significantly in WT mice (Figure 4C).

### 3.4. Cochlear Noise-Induced Inflammatory Response and Cell Death Are Exacerbated in Adult Igf1^+/−^ Mice

We studied the cochlear expression of factors involved in the inflammatory response in adult mice. Stimulation of the inflammatory response and macrophage recruitment following a noise challenge is commonly associated with cochlear injury. Indeed, pro-inflammatory cytokines *Tnfa*, *Tgfb1*, *Il1b*, or *Il6*, among others, are able to trigger apoptosis contributing to irreversible damage. In contrast, cochlear damage is restrained by induction of anti-inflammatory mediators such as FOXP3 or IL-10 that reduce the acute response [16,19,36,37]. These targets of noise-induced stress are also targeted by IGF-1, although causing the opposite effect [11,16]. *Tgfb1* expression was constitutively up-regulated in *Igf1*^+/−^ compared to WT mice (Figure 5A). *Tgfbr1* and *Tgfbr2* transcripts were further regulated by noise challenge in both genotypes, as reported for WT mice [35], with a distinct downregulation of *Tgfbr2* shortly after noise and sustained *Tgfbr1* expression in *Igf1*^+/−^ mice 3 days later. These mice also showed constitutive up-regulation of the expression of inflammatory cytokines and noise-challenge-induced time-dependent genotype-specific expression profiles of *Il1b*, *Il6*, and *Tnfa* (Figure 5B). *Il10* and *Foxp3* expression were down-regulated 4 h after noise in *Igf1*^+/−^ mice and increased 28 days later in WT mice. In both genotypes, *Dusp1* showed an increased expression 4 h after noise but recovered baseline levels 28 days later.

Three days after the noise challenge, IBA1^+^ macrophages infiltrated the lateral walls of the cochlea, confirming the exacerbated inflammatory response elicited in *Igf1*^+/−^ mice (Figure 5C; quantification and schematic drawing of the cochlea in Figure 5D).

TUNEL^+^ cells were abundant in the cochlear lateral wall (Figure 5C(d,h)). Four h post-noise challenge, c-Jun N-terminal kinases (JNK), but not p38, showed a two-fold activation in *Igf1*^+/−^ cochleae, and 28 days after noise, both increased (Figure 5E).

## 4. Discussion

As individuals age, there is a gradual decrease in IGF-1 and an increase in the generation of pro-oxidative and proinflammatory products that have an impact on tissue homeostasis and longevity [39,45,46]. In the mouse hearing organ, IGF-1 is essential for the final differentiation of the spiral ganglion and organ of Corti [38,43], and with aging, it maintains the stria vascularis physiology and auditory neuron survival. The works cited above also support the idea that IGF-1 is a predictive indicator of ARHL progression [39,47].

In this work, we showed that adult *Igf1*^+/−^ mice show normal hearing despite their having lower levels of plasma IGF-1 than WT mice. They showed no evident cellular alterations, but they did have altered cochlear gene expression. Following the noise challenge, the hearing of *Igf1*^+/−^ mice showed a progression worse than that of WT mice, suggesting a direct association between IGF-1 deficiency and susceptibility to NIHL. In younger mice, however, there were no differences between *Igf1*^+/−^ and WT cochleae, which could be attributed to the fact that IGF-1 levels were not yet under a “critical threshold”.

IGF-1 plays an important role in synaptic plasticity both in protecting sensorial synapses [24,47] and promoting cochlear synapse regeneration after excitotoxic trauma [48]. Adult *Igf1*^+/−^ mice showed altered auditory signal transmission. Thus, our data suggest that haploinsufficient mice may suffer silent synaptopathy that is further developed and uncovered by noise [49], similar to alterations in the auditory brainstem of the null mice previously described [50,51]. Furthermore, cochlear afferent innervation is also damaged by noise challenge and can be protected by efferent innervation [52].

Noise challenges induce several well-characterized morphological and functional changes in the cochlea, OHC being one of most affected cell types [53]. Increased NIHL observed in *Igf1*^+/−^ mice could be explained by the loss of HC found 28 days after the noise challenge test. IGF-1 has been reported to protect HC from ototoxicity by aminoglycosides through activation of the PI3K/AKT pathway in IHC and MEK/ERK pathways in the supporting cells surrounding OHC [54]. It has also been suggested that *Gap43* and *Ntn1* may be the possible effectors of IGF-1′s protective action [55]. Here, following a noise insult, we found no clear alterations in *Ntnt1* (data not shown), but *Gap43* expression was differently regulated in both genotypes by the noise challenge. These data are in agreement with the reported increased expression of this synaptic plasticity marker in the presence of IGF-1 [56] and also with its upregulation in the auditory brainstem following cochlear damage and HC loss [57].

IGF-1 signaling pathways were also altered in *Igf1*^+/−^ mice exposed to noise. Both AKT and ERK, which play protective roles in cochlear cells, were activated after a noise challenge [58,59]. However, the lack of AKT activation 28 days later could account for the death of HC caused by IGF-1 haploinsufficiency. JNK was also differently activated by noise in *Igf1*^+/−^ mice. JNK inhibition blocks noise-induced cell death pathways and, subsequently, NIHL [60]. In this work, we showed that adult *Igf1*^+/−^ mice present a lower antioxidant and inflammatory profile than WT mice. This might affect their capacity to face multiple insults, which constitutes one of the hallmarks of aging [61]. Principal pathogenic mechanisms in NIHL include oxidative stress [62] and the inflammatory response, with an early local expression of pro-inflammatory cytokines and recruitment of immune cells [63]. Free radical production and subsequent degradation of macromolecules in the cochlea also occurs in the hours following a noise challenge [64]. Here, we also reported time-dependent gene expression changes that, taken together, led to a favorable production of reduced glutathione in WT mice, thus suggesting a detour of intermediate metabolites toward a reinforcement of the antioxidant response [44]. This detour, however, was not observed in the adult *Igf1*^+/−^ mice exposed to the same noise insult.

The inner ear was considered an immune-privileged organ whose blood–labyrinth barrier was formed by the tight junctions between cells in the stria vascularis. However, various studies have identified resident cell types of macrophage lineage both in the stria and in the spiral ligament [65]. Immune cells increase in the cochlea after damage [66,67], and along with non-immune cells of the sensory epithelium, they express several immune and inflammatory genes such as *Tnfa*, *Il1b*, and *Il6* after acoustic overstimulation [68,69]. Moreover, anti-inflammatory cytokine IL-10 can be upregulated in the cochlea at early stages of the inflammatory response and its loss exacerbates SNHL [70]. In this study, we illustrated the differential responses shown by WT and *Igf1*^+/−^ mice after a noise challenge in the cytokine-mediated inflammatory response and the infiltration of the cochlea by immune system cells. In this sense, previous studies have shown that the age-related decrease in IGF-1 reduces the mouse lifespan and that it is associated with oxidative damage and a proinflammatory state [46]. In fact, IGF-1 regulates multiple aspects of immune cell function, and the factor itself is also regulated by products of the inflammatory “milieu,” underlying the interaction between the endocrine and immune systems [26,27]. Thus, proinflammatory cytokines decrease tissue sensitivity to IGF-1 inducing IGF-1 resistance and, in turn, IGF-1 decreases proinflammatory cytokines signaling, for example, by inducing IL-10 secretion [27,71,72]. IGF-1 is generally considered an anti-inflammatory factor and is especially involved in controlling the neuroinflammatory response associated with cerebrovascular pathologies [71,72] by regulating vascular permeability and modulating astrocyte response and the microglial phenotype, among other actions [25].

In summary, in this study we showed that IGF-1 haploinsufficiency increases susceptibility to noise injury and reduces the repair mechanism functions. This has important implications for understanding cochlear injury in aging individuals. Our data also support the notion that IGF-1 plays a central role in the maintenance of cochlear homeostasis and the regulation of inflammation, both of which are implicated in SNHL.

## Figures and Tables

**Figure 1 cells-10-01686-f001:**
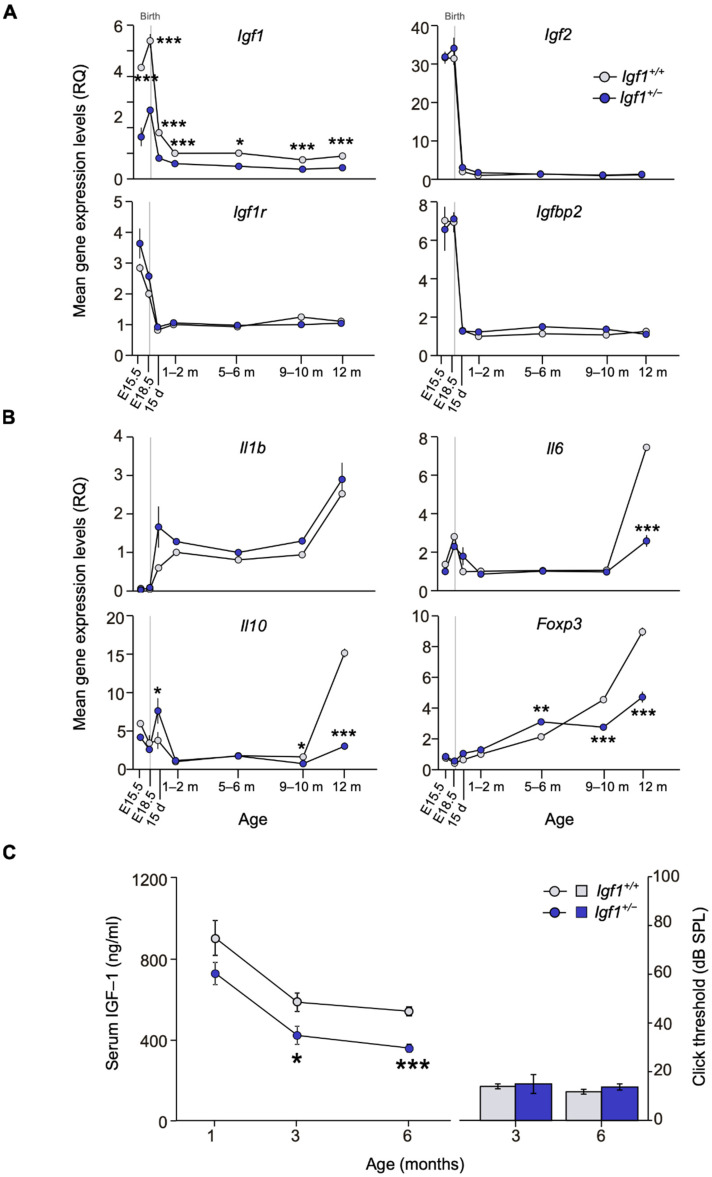
Evolution of cochlear gene expression and IGF-1 plasma levels in *Igf1*-heterozygous mice with age. (**A**,**B**) Cochlear gene expression of *Igf1*, *Igf2*, *Igf1r*, and *Igfbp2* (**A**) and of *Il1b*, *Il6*, *Il10*, and *Foxp3* (**B**) in WT and *Igf1*^+/−^ mice from embryonic (E) to adult stages. Expression levels were measured by RT-qPCR and calculated as 2^−ΔΔCt^ (RQ), using *Hprt1* as reference gene and normalized with the 1–2-month-old WT mice group. Values are presented as mean ± SEM of triplicates from pool samples of 3 mice per condition from two independent experiments. Statistically significant differences were analyzed by Student’s *t*-test (* *p* < 0.05, ** *p* < 0.01, *** *p* < 0.001 between genotypes). (**C**) IGF-1 serum levels (mean ± SEM of at least 4 mice per condition) were analyzed by ELISA in WT and *Igf1*^+/−^ mice in one-, three- and six-month-old mice. ABR thresholds (mean ± SEM) for three and six-month-old WT and *Igf1*^+/−^ mice did not vary and are shown for reference. Statistically significant differences were analyzed by Student’s *t*-test (* *p* < 0.05, *** *p* < 0.001). Days (d) and months (m).

**Figure 2 cells-10-01686-f002:**
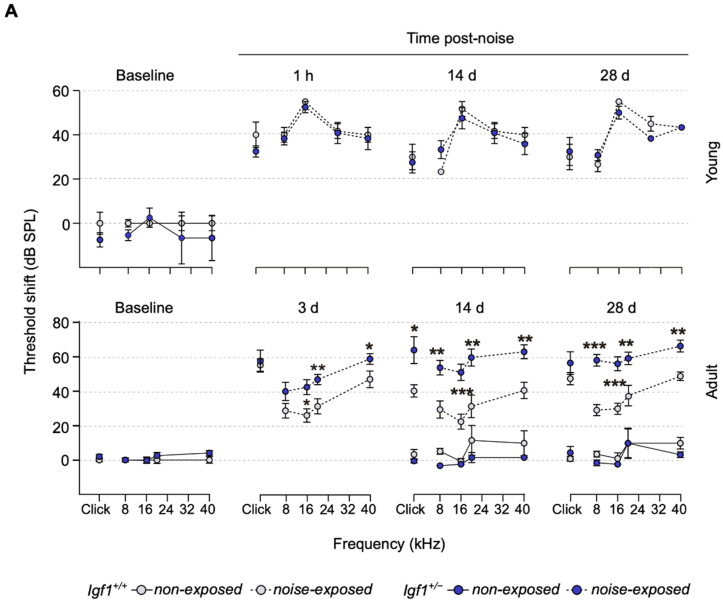
Differential response to noise damage in *Igf1*-heterozygous mice as they age. (**A**) Evolution of ABR threshold shifts (mean ± SEM) after noise exposure in young (n = 4 and 3) and adult (n = 9 and 7) *Igf1*^+/−^ and WT mice, respectively. ABR was evaluated before (baseline) and 1 h and 14 and 28 days post-noise exposure in young mice. In adult mice, ABR was evaluated before (baseline) and 3, 14, and 28 days post-noise exposure. Statistically significant differences were analyzed by Student’s *t*-test between genotypes (* *p* < 0.05, ** *p* < 0.01, *** *p* < 0.001). Electrical responses to broadband clicks and 8, 16, 20, 28, and 40 kHz pure tone stimuli, with an intensity range of 90–20 dB SPL in 5–10 dB steps, were recorded. (**B**) Representative ABR recordings in response to the click stimulus of adult WT and *Igf1*^+/−^ mice before (baseline) and 3 days after noise exposure; the solid black line marks the 10–20 dB SPL level above the auditory threshold where peak latencies were analyzed.

**Figure 3 cells-10-01686-f003:**
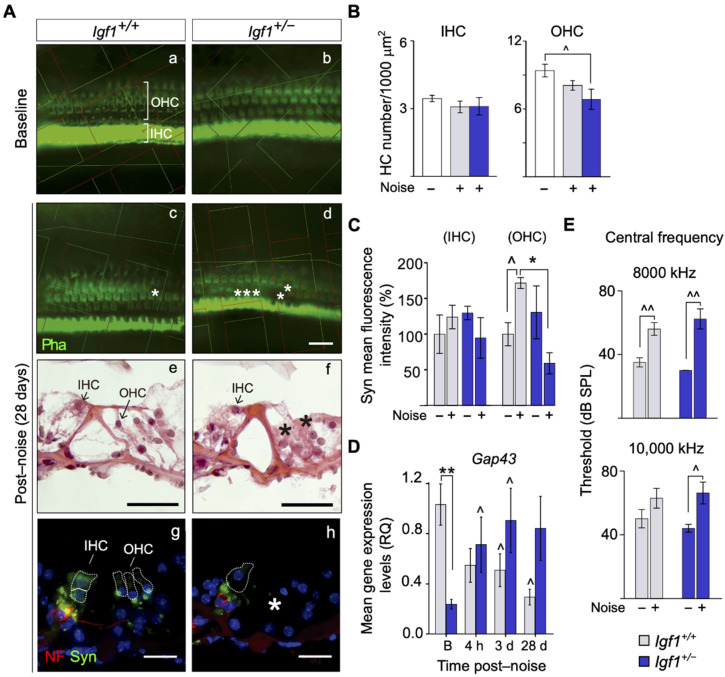
Cellular abnormalities and synaptic marker loss in the organ of Corti of *Igf1*^+/−^ mice after noise exposure. (**A**) Hair cell and synaptic marker loss in the organ of Corti 28 days after noise exposure. (**a**–**d**) Representative phalloidin (Pha)-stained samples in non-exposed and noise-exposed WT and *Igf1*^+/−^ mice. An unbiased counting frame grid was used for stereological counting that was photographed and is shown superimposed on each image. (**e**,**f**) Representative microphotographs of haematoxylin-eosin-stained cryosections of noise-exposed WT and *Igf1*^+/−^ mice. Asterisks indicate the absence of HC. (**g**,**h**) Representative microphotographs of cochlear cryosections showing the organ of Corti of noise-exposed WT and *Igf1*^+/−^ mice, 28 days after noise exposure. Sections were stained for neurofilaments (NF)—in red—and synaptophysi (Syn)—in green—showing at the base of HC nerve fibers and efferent synaptic terminals, respectively. Cellular nuclei were visualized with DAPI. The dashed lines show the approximate outline of selected IHC and OHC. White asterisk in h indicates the absence of synaptophysin. Scale bars: 50 μm. IHC: inner hair cell, OHC: outer hair cell. (**B**) Quantification of OHC and IHC in non-exposed and noise-exposed WT and *Igf1*^+/−^ mice. Values are presented as mean ± SEM of at least 3 mice per condition. Statistically significant differences were evaluated by the Mann–Whitney U test. (^ *p* < 0.05 vs. baseline condition.) (**C**) The intensity of synaptophysin staining was quantified using ImageJ software. Data were obtained from 11 to 20 sections of at least 3 mice from each condition and are shown relative to those of non-exposed WT mice as mean ± SEM. Statistically significant differences were evaluated by the Mann–Whitney U test (* *p* < 0.05 between genotypes; ^ *p* < 0.05 vs. baseline condition). (**D**) *Gap43* cochlear mRNA expression analyzed by RT-qPCR in WT and *Igf1*^+/−^ before (**B**) and 4 h and 3 and 28 days after noise exposure. Gene expression levels were calculated as 2^−ΔΔCt^ (RQ) normalized with data from baseline WT mice group. Values are presented as mean ± SEM of at least 3 mice per condition evaluated in triplicate. Statistically significant differences were evaluated by the Mann–Whitney U test (** *p* < 0.01 between genotypes; ^ *p* < 0.05 vs. baseline group). (**E**) DPOAE threshold (mean ± SEM of at least 4 mice per condition) in non-exposed and noise-exposed WT and *Igf1*^+/−^ mice 28 days after noise exposure. Statistically significant differences were evaluated by the Mann–Whitney U test (^ *p* < 0.05; ^^ *p* < 0.01 vs. baseline condition). (**F**) Cochlear protein relative levels were measured by Western blotting. Representative blots and quantification of levels are shown for p-AKT, p-ERK1/2 cochlear protein extracts 4 h and 28 days post-noise exposure in noise-exposed and non-exposed WT and *Igf1*^+/−^ mice. Expression levels were calculated as a ratio using PI3K or non-phosphorylated forms of AKT and ERK and normalized to non-exposed WT mice group. Values are presented as mean ± SEM of at least 3 mice per condition. Statistically significant differences were evaluated by the Mann–Whitney U test (* *p* < 0.05 between genotypes; ^ *p* < 0.05 vs. baseline group).

**Figure 4 cells-10-01686-f004:**
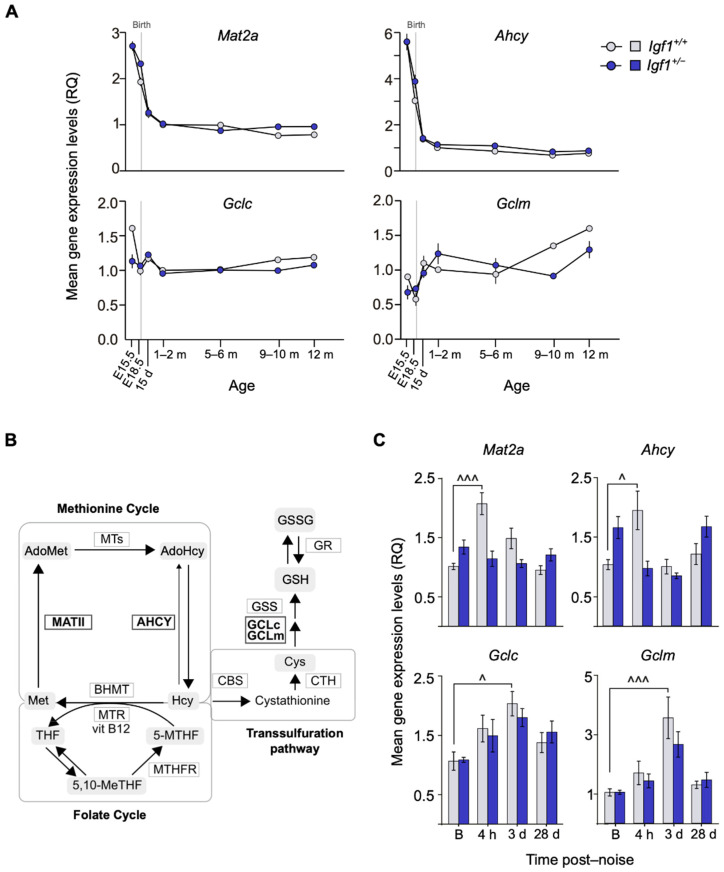
Comparative mRNA expression of oxidative-stress-related genes in noise-exposed *Igf1*^+/−^ cochlea. (**A**) Cochlear gene expression of *Mat2a*, *Ahcy*, *Gclc*, and *Gclm* in WT and *Igf1*^+/−^ mice from embryonic (E) to adult stages. Expression levels were measured by RT-qPCR and calculated as 2^−ΔΔCt^ (RQ), using *Hprt1* as a reference gene and normalized with data from the 1–2-month-old WT mice group. Values are presented as mean ± SEM of triplicates from pool samples of 3 mice per condition from two independent experiments. Statistically significant differences were analyzed by Student’s *t*-test. Days (d) and months (m). (**B**) Schematic representation of homocysteine metabolism and its intersection with glutathione (GSH) synthesis and folate cycle including the enzymes analyzed in this work (marked in bold). Adapted from Partearroyo et al. [44]. (**C**) Cochlear mRNA expression levels of *Mat2a*, *Ahcy*, *Gclc*, and *Gclm* analyzed by RT-qPCR in WT and *Igf1*^+/−^ mice before (**B**) and 4 h and 3 and 28 days after noise exposure. Gene expression levels were calculated as 2^−ΔΔCt^ (RQ) normalized with data from the baseline WT mice group. Values are presented as mean ± SEM of at least 3 mice per condition evaluated in triplicate. Statistically significant differences were evaluated by one-way ANOVA and the post-hoc Bonferroni’s multiple comparison test (^ *p* < 0.05; ^^^ *p* < 0.001 vs. baseline group).

**Figure 5 cells-10-01686-f005:**
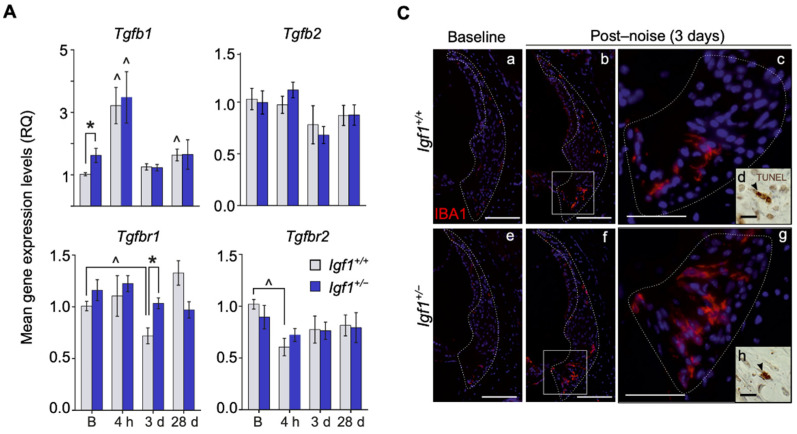
Noise effects on the activation levels of inflammation-related genes and proteins, infiltration of immune cells, and apoptosis-related genes in the cochlea of *Igf1*^+/−^ mice. (**A**,**B**) Cochlear mRNA expression levels of (**A**) *Tgfβ1*, *Tgfβ2*, *Tgfβr1*, and *Tgfβr2* and (**B**) *Il1β*, *Il6*, *Tnfα*, *Il10*, *Foxp3*, and *Dusp1* were analyzed by RT-qPCR in WT and *Igf1*^+/−^ mice before (**B**) and 4 h and 3 and 28 days after noise exposure. Gene expression levels were calculated as 2^−ΔΔCt^ (RQ) normalized with respect to the baseline WT mice group. Values are presented as mean ± SEM of at least 3 mice per condition evaluated in triplicate. Statistically significant differences were evaluated by the Mann–Whitney U test (* *p* < 0.05 between genotypes; ^ *p* < 0.05; ^^ *p* < 0.01 vs. baseline group). (**C**) Representative microphotographs of cochlear mid-modiolar cryosections immunolabeled for IBA1 (red) showing (**a**–**g**) the spiral ligament of the cochlea in noise-exposed and non-exposed WT and *Igf1*^+/−^ mice 3 days after noise exposure. Close-ups of the region of type IV fibrocytes in the spiral ligament are shown in (**c**,**g**). TUNEL positive cells were also found in this area (arrowheads in **d** and **h**). Scale bars: (**a**,**b**,**e**,**f**) 100 μm, (**c**,**g**) 50 μm, and (**d**,**h**)10 μm. (**D**) IBA1 total fluorescence intensity was measured in the spiral ligament using ImageJ software. Data were obtained from 14 to 23 sections of at least 3 mice from each condition and are shown relative to those of non-exposed WT mice as mean ± SEM. Statistically significant differences were evaluated by the Mann–Whitney U test (^ *p* < 0.05 vs. baseline group). Schematic drawing showing the cross section of the cochlea. SM: scala media, ST, scala timpani, SV scala vestibuli, SG: spiral ganglion, OC: organ of Corti, LW: lateral wall, SV: stria vascular, SPL: spiral ligament. (**E**) Cochlear protein relative levels were measured by Western blotting. Representative blots and quantification of levels are shown for p-p38 and p-JNK cochlear protein extracts 4 h and 28 days post-noise exposure in noise-exposed and non-exposed WT and *Igf1*^+/−^ mice. Expression levels were calculated as a ratio using PI3K or the non-phosphorylated forms of p38 and JNK and normalized to non-exposed WT mice groups. Values are presented as mean ± SEM of at least 3 mice per condition. Statistically significant differences were evaluated by the Mann–Whitney U test (* *p* < 0.05 between genotypes; ^ *p* < 0.05 vs. baseline condition).

## Data Availability

Data are available upon request to authors and will be available on the CSIC open data institutional repository.

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
