# Peer review of "IGF-1 Haploinsufficiency Causes Age-Related Chronic Cochlear Inflammation and Increases Noise-Induced Hearing Loss"

_cells, 2021, doi:10.3390/cells10071686_

Round 1
Reviewer 1 Report
This manuscript well documents the effects of IGF1 haploinsufficiency on age-related degeneration of mouse cochleae. Results are consistent with previous findings in the effects of IGF1 insufficiency on cochlear damage. Actually, this study provides the molecular bases for hearing loss due to IGF1 deficiency, which is suitable for publication in Cells.
There are a few concerns before publication.
1) Images of figure 3A are unclear. In addition, several square lines are overlaid on fluorescence images, which should be deleted. In methods, the authors described that hair cells were identified by intact hair bundles labeled by phalloidin. The quality of these images should be matched to these criteria.
2) In methods, there is no description for immunostaining. The list of the primary antibodies is to be represented.
3) In figure legends, abbreviations that appear in figures should be explained. For instance, NF and Syn in figure 3Ag are difficult to understand. In addition, the description of figure 3Aa is an unusual style.
4) The reason for use of synaptophysin as a marker for synapses should be described. In general, the primary cite of damages in cochlear synapses after noise exposure are found in afferent synapses at IHCs, and estimation of numbers in ribbon synapses is frequently used.
5) In figure 5, the expression levels of several inflammatory cytokine mRNAs are demonstrated. There is no description of the reason why the authors chose those molecules to be evaluated, which should be stated. In addition, their hypothesis for roles for those cytokines and possible pathways should be demonstrated as a schema like Figure 4B.
Author Response
Author´s answers to Reviewer 1
We greatly appreciate the reviewer´s comments that have improved the quality of the manuscript. We have used a stereological approach to define a cochleogram and estimate HC density, as reported by others and by us (34). The area of interest containing the rows of HC was divided into equidistant 5% sectors using CAST® stereological software (v.2.3.2.0, Visiopharm) in an Olympus BX51 microscope connected to an Olympus DP70 video camera with a source of ultraviolet light (Olympus U-RFL-T). Microphotographs shown are the best quality possibly by using this system and were taken on organ of Corti whole mounts with the overlaid grid, which cannot be deleted (comments 1 and 4). The revised manuscript includes a detailed description of the methods used as requested in comments 1, 2, 3 and 4. Figure legends have also been modified as requested (comment 3). The list of antibodies used was presented in Supplementary Table 1 (comment 2). NF and Syn expression alterations were associated to homozygous Igf1 deletion in mice, therefore we tested if these alterations were also present in the heterozygous Igf1 mouse. This is now clearly stated and referenced in the text (page 6, comment 4). Finally, we specially appreciate comment 5 by this reviewer; the reason underlying the study of this set of cytokines is now clearly explained and referenced in the revised text (page 9, comment 5). Figure 4 is already complex and there is no simple scheme to explain the interaction among cytokines, thus we have chosen to include recent reviews in the list of references that could help the reader to better understand the rationale behind these experiments.
Reviewer 2 Report
In this study, the authors show that Igf1 heterozygous mice after noise exposure presented increased expression of inflammatory cytokines, greater susceptibility to injury hair cells, moreover reduces the repair mechanism functions. This work its important because contributes for elucidation of the molecular bases to age-related hearing loss linked to IGF-1 deficiency.
Review suggestions:
Item 2, page 2: Materials and Methods
- Which mouse strain was used? It is important to note that some strains are more susceptible to age-related hearing loss.
- Were males and females used?
- Was the hearing loss induced for only 30 minutes?
- In this item, detail how often the animals' hearing was assessed - before and after exposure to noise.
Results
- Page 6, Line 181 it writes “....24 hours after noise challenge, young mice (Figure 2A, upper panel) of both genotypes presented similar 40 dB threshold shifts and recovery profiles”; however in figure 2A it is indicated 1h and not 24h! It is important to rule out the periods in which hearing was assessed among the studied groups.
- Figure 3, page 8: identify in Figure C that it is the mean of the fluorescence intensity of the protein marker and in Figure D that it is the mean expression of mRNA.
- Identify in figures 4 (page 10) and 5 (page) when they are gene expression/transcript and when they are proteins, for example: RQ gene
Author Response
Author´s answers to Reviewer 2
We appreciate very much the careful revision and positive comments by this reviewer. We have revised the Methods section to include more details on the aspects requested. Both male and female MF1/129SvEvTac mice were used, since we observed no differences data were pooled. Mice were exposed to noise for 30 min in an in-house patented chamber by following a procedure extensively reported by us; the procedure is now described in full detail under the Methods section. ABR thresholds were measured before and 1 h, 3, 14 and 28 days after noise exposure. ABR and DPOAE procedures are now better explained in the revised text. The description of Figure 2A has been corrected, and Figures 3, 4 and 5 have been revised as suggested by the reviewer.